# CONTRASTIVE LEARNING WITH 3D SHAPES

**Andrea Bernini**
Department of Computer Science
La Sapienza University
Rome, Italy
`bernini.2021867@uniroma1.studenti.it`

## ABSTRACT

In fields such as Computer Vision or NLP, there is a large amount of data available which, however, cannot be labeled, as it would be very expensive. A possible solution to this problem is Contrastive Learning, a Self-Supervised technique. This work aims to implement a contrastive learning regime for a non-Euclidean data type, more precisely 3D Point Cloud Shape.

## 1 INTRODUCTION

Contrastive Learning is a machine learning technique used to learn the general characteristics of an unlabeled dataset by teaching the model which data points are similar or different by incorporating versions of the same sample next to each other while attempting to push away the embeds from different samples (as explained in "Understanding Contrastive Learning (Tiu, 2021)"). This technique is very useful because manually annotating unlabeled datasets is a very time and money-consuming task, especially in fields such as *Computer Vision* or *Natural Language Processing*, where there is an increasing amount of data that is not labeled. The Contrastive Learning pipeline can be divided into three main sections:

- **Augmentation**: for each sample in our dataset, two combinations of augmentation are performed, in such a way as to use the original image as an anchor, its augmented version as a positive sample, and the rest of the images in the batch (or in the training data) as negative samples.

- **Encoding**: we feed our augmented data into our deep learning model, to create vector representations for each sample. The goal is to train the model to produce similar representations for similar samples.

- **Training**: we try to maximize the similarity of the two vector representations by minimizing a *Contrastive Loss* function.

## 2 METHOD

**Dataset**   The *PyTorch Geometric* library contains many standard benchmark datasets regarding 3D figures. We decided to use *Shapenet*, a dataset containing 3D shape point clouds from 16 shape categories because thanks to its parameters it is possible to select only a subset of categories of figures, which is very useful to avoid problems related to the occupied space in RAM.

**Model**   There are several possible approaches to managing the 3D point cloud, such as *Voxelization*, which is not very efficient, or *Multi-View*, which is more powerful than the first, but we don't use 3D information. However, there are Deep Learning models that can operate directly on it, thanks to the fact that they are invariant to permutations and therefore to the ordering of points. Examples of these models are: *PPFNet* (Birdal et al., 2018), *PointNet/Pointnet++* (Guibas et al., 2007), *EdgeConv/DynamicEdgeConv* (Bronstein et al., 2020).

For the network architecture, we took inspiration from the SimCLR (for more information see "A Simple Framework for Contrastive Learning of Visual Representations (Chen et al., 2020a).", where an **End-to-End** architecture is used (Figure 1), i.e. for each sample $x$, we have two augmented

| Model | (Max 10 values) Mean Accuracy | (Min 10 values) Mean Train Loss |
|---|---|---|
| Contrastive DS | 52.63 % | 2.13 |
| Classifier | 98.85 % | 0.39 |

Table 1: Performance Evaluation, between Contrastive Downstream Task and Simple Classifier.

versions ($\tilde{x}_i$ and $\tilde{x}_j$), which we will pass to the encoder $f(\cdot)$ (consisting of two **DynamicEdgeConv** layers), to obtain a representation vector. Then we pass this to the Projection Head, a fully connected network. The projection head $g(\cdot)$ maps the representation $h$ into a smaller space where we apply the loss function.

The reason why it is necessary to apply the Projection Head is that most similarity measures that are used in contrastive losses suffer from the *curse of dimensionality*. Therefore, when we calculate the loss on smaller vectors, we will get better results (as we can see in "*A Simple Framework for Contrastive Learning of Visual Representations* Chen et al. (2020b).").

**Loss Function**  As Loss function we used **NTXentLoss**, which compare the similarity of $z_i = g(h_i)$ and $z_j = g(h_j)$ to the similarity of $z_i$ to any other representation in the batch $z_k$, by performing a *soft-max* over the similarity values. The loss can be formally written as:

$$\mathcal{L}_{i,j} = -\log \frac{\exp(\mathrm{sim}(z_i, z_j)/\tau)}{\sum_{k=1}^{2N} 1\!\!1_{[k \neq i]} \exp(\mathrm{sim}(z_i, z_k)/\tau)} \tag{1}$$

where $\tau$ is the *temperature*, $1\!\!1_{[k \neq i]}$ is an indicator function (1 if $k \neq i$, 0 otherwise), and $sim()$ is the application of Cosine Similarity. The code of this work is available at the following web address: `https://bit.ly/iclr_contrastive_learning`

## 3 EXPERIMENTAL RESULTS

For testing, we used a **Downstream** approach (as SimCLR) to see how the model performs with small data. We used logistic regression to see if the model generalizes well, and we'll pass the 3D elements that have already been processed by our $f(\cdot)$ function, i.e. encoded in their feature vector, to associate the representations with a class prediction. By doing this, we have reached an average accuracy level of 51% for the first 10 highest values, in 100 epochs. To have a point of reference to compare the results obtained with Contrastive Learning, I decided to train another network with the same architecture as the previous one (therefore two DynamicEdgeConv layers and the projection head with a fully connected network), with the same hyperparameters, and with the same optimizer, but using it as a classifier, thus exploiting the database labels. With this model, we obtain far superior results compared to learning by contrast, reaching 98.85% average accuracy among the ten highest values, in 100 epochs. In Figure 2, we can see the accuracy and train loss of the two models compared, while in Table 1, we have a summary of the performance of the two different approaches.

## 4 DISCUSSION AND CONCLUSION

One of the main problems is the use of the End-to-End architecture, which, as explained in *A survey on contrastive self-supervised learning* Banerjee et al. (2020), works well only in the presence of a large number of negative samples and, therefore, a large batch size. In fact, due to hardware limitations, We were only able to use batches of 64 elements. A future improvement could be to use a different architecture that has no problems related to the use of the GPU memory, for example, the Memory Bank or the Momentum Encoder.

URM STATEMENT

Author **Andrea Bernini** meets the URM criteria of ICLR 2023 Tiny Papers Track.

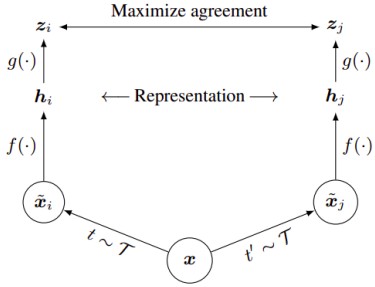

Figure 1: Contrasting Learning model architecture (Chen et al., 2020a).

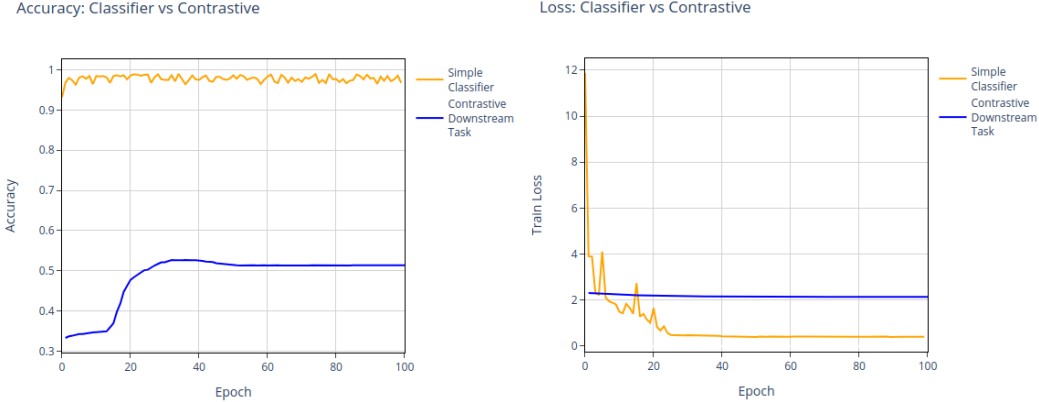

Figure 2: Comparison between Contrastive Downstream Task and Simple Classifier.

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
