# OpenReview forum: "Contrastive Learning with 3D Shapes"
_ICLR.cc/2023/TinyPapers — Submitted to Tiny Papers @ ICLR 2023_

### Official Review · Reviewer_Cij7 · 2023-03-26

**Confidence:** 4

**Summary Of Contributions:**

The paper proposes using a SimCLR-like approach for 3D point cloud representation learning. Preliminary results are shown using the ShapeNet dataset.

**Rating:**

Great Start (GS): a submission which meets some of the reviewing criteria but has room for improvement

**Strengths And Weaknesses:**

Strengths
* The motivation of the paper is good: labels are scarce in reality and we need to explore better methods for self-supervised learning
* it is great that the authors share their code: transparency and reproducibility is very important in our field

Weaknesses
* The paper writing could be done with more care. Fig 1 is copied from the SimCLR paper: could it instead by augmented with the differences between this work and that? SimCLR was tested on images: what has been changed here to allow the method to be used on point clouds?
* It is unclear what the task is here. 3D point cloud shape is mentioned in the abstract but there is no data mentioned. Although the details may exist in the code, they must be described (at least summarized) clearly in the paper - readers are not expected to refer to code to get the high-level details of what is going on.
* The results should be framed more clearly: why are the numbers interesting? How does the result change the understanding of the problem, or show the promise of the method? Achieving 51% accuracy in 100 epochs might be really great or it might not be, the reader cannot tell without further information.

This is neither a strength nor a weakness but I definitely appreciate the struggle with batch sizes. It is difficult to use contrastive methods like SimCLR which rely on the availability of significant hardware to compute. I admire that the authors are exploring these methods despite the lack of access to compute. However, I would also encourage consideration of research problems which better suit the available resources. I remember as a PhD student seeing the papers coming out of large industrial labs and feeling a bit demoralized, that I wanted to do what they were doing but did not have the resources. A way to avoid this feeling, until you one day join one of those labs, is to choose problems which require more nimble thinking rather than large compute (e.g. start from papers which you know you can replicate on your machine); as this, I think, can be the strength of an under-resourced researcher :)

**Suggested Changes:**

Unfortunately it is quite difficult to gauge the contributions of the work from the paper alone. I encourage the authors to take into consideration the feedback above and to please have another go soon. Here are also some style tips below which help to improve the readability of the work:

* For citations, there is no need to write the title of the paper in the main text as readers can check the bibliography
* This avoids making mistakes like ”A Simple Framework for Contrastive Learning of Visual Representations Chen et al. (2020b).”
* Avoid "I" even for a single-author paper - always use "We"
* Open quotations with `` rather than ''
* down-stream -> downstream

---

### Official Review · Reviewer_txG6 · 2023-03-27

**Confidence:** 4

**Summary Of Contributions:**

This paper tries to tackle the problem of freely available unlabelled data using self-supervised learning to learn useful representations. In particular, this paper discusses the use of Contrastive Learning for 3D Shapes.

**Rating:**

Great Start (GS): a submission which meets some of the reviewing criteria but has room for improvement

**Strengths And Weaknesses:**

### Strengths :
-  The paper is very well written and easy to follow. The brief introduction to Contrastive Learning and their method is really well done.
-  The inclusion of code makes the results seem more credible (although reproducing the paper is also not a hard thing to do)
-  The discussion section dives into nice future directions that the author themselves can try (or anyone reading the paper)

### Weaknesses :
-  The Experimental Results section is inadequate. I would rather see a small table with some more results compared to 2 curves.
-  I am not sure what kind of data set you used in the experimental results, or some more architecture details for the replication of the same result like the projection head.


**Suggested Changes:**

-  I think it would be nice to see some more experiments on different data sets (maybe like 1 or 2 more).
-  I do not currently see a baseline in the paper. I am sure there are some works out there that work with SSL + Point clouds, so maybe in the table of results you could add them so the reviewer can compare how your approach fairs in comparison to other such approaches. It is not a bad thing to include baseline that at first might outperform your method given that the paper is not meant to be too much detailed, and slowly but surely work your way to improve your method compared to the base lines !

---

### Author Response · Authors · 2023-05-30
**Opt-in for archival**

I want to opt for archiving.

---

### Comment · Area_Chair_hJi1 · 2023-06-06
**Meets threshold for archival**

This work meets the threshold for archival, contents the URM statement and is deanonymized

---

### Meta-Review · Area_Chair_hJi1 · 2023-04-09

**Recommendation:** Invite to archive
**Confidence:** 3

**Metareview:**

The paper presents a contrastive learning methodology that is suitable for 3d shapes. The authors adapt the SimCLR framework for contrastive learning along with the recent deep learning model developments that are invariant to permutations, and thus suitable for point clouds.

The reviewers appreciate how well written the paper is. It provides a good introduction to contrastive learning in general, along with a brief introduction to methods for point clouds. Their methodology is clear and easily reproducible (code provided).

However, the reviewers agree that the results section is insufficient. No baseline model or other context is given for the reported accuracy. Furthermore, the paper does not indicate any details about the data used.

Overall, the paper presents a compelling methodology for an important class of data. Pending further experiments and an expanded discussion of the results, this work appears to be very promising.

**Summary:**

This paper explores contrastive learning for 3D shapes by adapting SimCLR for point clouds.

**Reason For Not Giving A Higher Recommendation:**

The results section is inadequate.

**Reason For Not Giving A Lower Recommendation:**

The paper is well written and is both clear and reproducible.

---

### Decision · Program_Chairs · 2023-04-10

Invite to archive